# Channelized Substrates Made from BaZr_0.75_Ce_0.05_Y_0.2_O_3−d_ Proton-Conducting Ceramic Polymer Clay

**DOI:** 10.3390/membranes9100130

**Published:** 2019-10-09

**Authors:** Sandrine Ricote, Benjamin L. Kee, W. Grover Coors

**Affiliations:** 1Department of Mechanical Engineering, Colorado School of Mines, 1500 Illinois street, Golden, CO 80401, USA; 2Hydrogen Helix, PO Box 16070, Golden, CO 80402, USA; grover.h2helix@icloud.com

**Keywords:** proton-conducting ceramic, fabrication, BZCY, polymer clay, solid-state reactive sintering

## Abstract

A novel process for producing thick protonic ceramics for use in hydrogen separation membrane reactors is demonstrated. Polymer clay bodies based on polyvinyl acetate (PVA) and mineral oil were formulated, and they permitted parts with complex architectures to be prepared by simple, low-pressure molding in the unfired, “green” state. Ceramic proton conductors based on doped barium zirconate/cerate, made by solid-state reactive sintering, are particularly well-suited for the polymer clay process. In this work, the ceramic proton conductor, BZCY755 (BaZr_0.75_Ce_0.05_Y_0.2_O_3−d_) was fabricated into a variety of shapes and sizes. Test coupons were produced to confirm that the polymer clay route leads to a high-quality ceramic material suitable for the demanding environment of high-temperature membrane reactors. It has been demonstrated that protonic ceramic specimens with the requisite properties are easily prepared at the laboratory scale. The polymer clay fabrication route opens up the possibility of high-volume, low-cost manufacturing at a commercial scale, by a process similar to how dinnerware and sanitary porcelain are produced today.

## 1. Introduction

Many energy conversion devices require flow channels, where a working fluid—whether gas or liquid-exchanges some thermodynamic quantity with the surroundings [1,2,3]. It is necessary to ensure that no gas leaks out of these channels along the way between the inlet and the outlet. At low temperatures, the flow channels are typically sealed using elastomer gaskets. At higher temperatures, mechanical compression seals are often employed, but these introduce added system complexity and risk due to gasket failure. The possible sealing methods for ceramic devices requiring extended operation at temperatures above about 500 °C include glass-ceramic bonding, ceramic-metal brazing, and ceramic-ceramic bonding [4,5]. Glass–ceramic and ceramic-metal brazing are convenient because they enable the joining of pre-sintered and machined ceramic components. Ceramic-ceramic bonding is preferred because it is easier to manage the coefficient of thermal expansion difference across joints, however, this type of joining requires that different ceramic components bond together during sintering. A device, where the flow channel is fully enclosed by one or more hermetic ceramic components joined during sintering, is the basis of an all-ceramic channelized substrate.

The three-dimensional channel architectures are typically fabricated in ceramic components in the ‘green’ state prior to sintering. The ceramic powders are combined with organic binders to permit forming into the desired shape [6]. For devices with uniform radial profiles—tubes, for example—channels may be produced by the extrusion of ceramic bodies through dies under high pressure. For planar devices, the channel structures are typically made by green machining of a plate, dry-pressing in a die from powders, pressed from soft pre-formed tape, or stamping the various layers from cast tape and subsequently laminating of the various layers under heat and pressure. More exotic methods for shaping green bodies exist, such as thermoplastic injection molding and 3D additive printing. Each process has particular advantages and disadvantages, capital equipment requirements, and scalability.

Ceramic bodies that flow in the transverse direction to the application of the force are called ‘plastic’ when the deformation remains. Compaction of powders under pressure by dry-pressing does not permit material to flow laterally in the die, so pressing channels and other features require complex dies. Plastic bodies, on the other hand, lend themselves to low-cost ram pressing at much lower applied pressures [6,7]. Plastic bodies called polymer clay are produced by combining ceramic powders with a suitable viscous binder to provide green strength after drying. Polyvinyl acetate (PVA) is a good choice because the clay is exceedingly plastic prior to drying, and hard and durable after drying. Water is added to adjust clay viscosity, and mineral oil is added to serve as a lubricant for flow and to prevent sticking, in much the same way that butter is used in making pastries.

Channelized substrates for ceramic electrochemical devices, such as fuel cells, electrolyzers, and membrane reactors/separators, generally use porous ceramic electrodes containing nickel oxide [8,9,10,11,12,13], which is an inhalation hazard in powder form. One of the principle advantages of ram pressing channelized substrates from polymer clay bodies is that the exposure to NiO dust is minimized. Once the NiO powder is incorporated into the clay body, NiO powder becomes encapsulated in the binder, so the bodies may be handled safely during processing without the need for dust collection systems. Also, ram pressing permits parts to be fabricated to near-net shape, with minimal waste. This is in contrast to green machining, where material removed from the channels and exterior perimeter must be handled as hazardous waste or recycled.

Finally, complex and fully functional ceramic specimens may be produced at small scale using simple tools. Batch sizes on the order of 10 g are routine with ram pressing from polymer clay, in contrast with kilograms required for nearly all other processes.

For millennia, ceramic objects have been formed from clay because this process is simple and inexpensive to practice, from the home potter’s wheel to huge factories making dinnerware and sanitary porcelain. In the age of advanced ceramics, however, green forming from plastic bodies has been largely supplanted by technologies that permit higher packing densities of the refractory powders, making it possible to obtain higher fired density than may be achieved at lower sintering temperatures, which are a requirement for most high-purity industrial ceramic materials. Yttrium-doped barium zirconate/cerate (BaZr_1−x−y_Ce_x_Y_y_O_3−d_, BZCY) ceramics are used as high-temperature proton conductors in fuel cells and hydrogen separation membranes [8,9,10,11,12,13,14,15,16,17,18,19]. This is a particularly demanding application because the ceramic must be fully dense, with no open porosity that would permit leakage of hydrogen gas. The traditional method for making these ceramics consists of preparing nanopowders of phase-pure materials, and compacting them in the green state. This method does not work very well, because barium zirconate is a very refractory material that resists grain boundary formation at reasonable sintering temperatures. The discovery of solid-state reactive sintering (SSRS) has proven to be an effective method for making dense protonic ceramic membranes with large grains and high proton conductivity [11,15,20,21,22,23,24]. During sintering, the perovskite protonic ceramic phase is formed in-situ from the precursor powders, BaCO_3_, ZrO_2_, CeO_2_ and Y_2_O_3_, mixed in the correct stoichiometry. This is made possible by the addition of a small amount of NiO, which reacts with BaCO_3_ to form a eutectic glass phase during the onset of sintering. The glassy phase promotes decomposition of the oxides and solid-state reaction of the components. Barium is extracted from the NiO–BaO glass during sintering to form the BZCY phase, leaving a small residue of NiO behind.

One of the consequences of SSRS is that the high green densities required for sintering pre-reacted BZCY powder are not necessary, because solid-state reaction and sintering occur simultaneously. With this process, it is not necessary to limit the amount of organic binder and rely on high compaction pressures. With SSRS, a large window exists for using organic polymers as binders, making plastic bodies possible that can be formed using many of the same methods that are used in forming clay bodies. Polymer clay processes have been employed for a long time for making low-density ceramic parts. However, high-density parts, suitable for hydrogen separation membranes can be fabricated by SSRS, including intricate shapes molded at low pressure. With SSRS, the dense pore-free microstructure required for hydrogen separation is produced during sintering so that high packing density of the ceramic powders is not required. This makes the use of powders that are highly filled with organic binders more feasible—a necessary requirement for polymer clay.

## 2. Materials and Methods 

BaZr_1−x−y_Ce_x_Y_y_O_3−d_ ceramic proton conductors have the general ABO_3_ perovskite structure. In addition, NiO is blended in with the other premixed powders. The formulation that has been employed in this work is BaZr_0.75_Ce_0.05_Y_0.2_O_3−d_, referred to as BZCY755. The precursor powders were blended with 0.5 wt% NiO to make electrolyte membranes or with 50 wt% NiO to make ceramic-metal (cermet) electrodes.

### 2.1. Polymer Clay Formulation

The inorganic ceramic powders, called the precursors, were prepared according to stoichiometry: 61.5 wt% BaCO_3_ (Alfa Aesar 14341), 28.8 wt% ZrO_2_ (Alfa Aesar 40140), 2.7 wt% CeO_2_ (Alfa Aesar 11328), and 7.0 wt% Y_2_O_3_ (Alfa Aesar 11180). 0.5 wt% NiO (Alfa Aesar 45094) was introduced as the reactive sintering additive. The precursor powders, with a batch size totaling 500 g, were milled in acetone in a jar roller for 24 h with YSZ media and pan dried. Smaller batches were dry-mixed in a mortar and pestle.

Every powder has a different surface area and particle size distribution. In principle it is possible to calculate the open pore volume from the packing fraction of the powders, but in the end, much trial and error was required to arrive at a binder formulation giving the right clay-like properties. The binder system consisted of mineral oil, as a lubricant, and PVA glue as the binder/plasticizer. The polymer clay was a water-based system, and water was removed by drying. The non-ceramic ingredients must burn out cleanly during sintering. The binder used was ordinary PVA glue, with a weight loss on drying (LOD) of 50%. Common food-grade mineral oil (Up and Up) was used as the lubricant. Deionized water was added to adjust rheology. The amount of water determines the clay-like properties—too much water, and the clay is too soft and sticky, and too little water makes the clay brittle with poor flow characteristics. Wood glue has just about the right balance between water and PVA to give the clay body the proper forming characteristics with the addition of a small amount of extra water.

50 to 100 g batches of dried precursor powder were introduced into a food processor (Cuisinart Mini-Prep Food Chopper, model DLC-1SS) with 7 wt% mineral oil and 18 wt% deionized water. The ingredients were blended three times with material scraped off the walls of the blender with a plastic scraper each time to ensure uniform dispersion of the oil and water throughout the powder matrix. The powder prepared in this way resulted in a “crumb-like” matrix suitable for transfer into an airtight container for later use. Stored this way, the precursor blend has a shelf-life of weeks.

Batches of polymer clay, consisting of 10 to 20 g of the precursor blend, were prepared by introducing 20 wt% PVA glue (Titebond Original Wood Glue) to the precursor blend (containing oil and water) and stirring with a polyethylene spatula, followed by rigorous kneading on a non-stick pastry sheet until the clay achieved a uniform, clay-like texture. Water was added as required to adjust the clay rheology from batch to batch. Kneading forces out the oil, preventing sticking. The polymer clay was either molded immediately, or placed in an airtight container or plastic bag for later use. As the surface of the clay begins to dry out, it quickly loses its ability to bond to itself and cracks when molded. If the clay is too moist, it becomes sticky and hard to mold. The glue dries quickly so the working time of the polymer is only a few minutes before good molding properties are lost.

### 2.2. Specimen Forming

Several types of parts were fabricated from BZCY755 electrolyte as well as BZCY755/NiO electrode compositions: discs, rods and molded channelized substrates. Thin discs were fabricated from softer clay (i.e., slightly more water) by placing a ball of clay between two sheets of clear plastic film (e.g., 0.1 mm thick poly report covers). The clay was then compressed between two glass plates to the desired diameter and thickness using plastic shims. Polymer clay sheets as thin as 200 microns were prepared in this way. Rods were prepared by rolling the clay between a glass plate and the pastry sheet. Shims were attached to the glass plate to set the finished diameter. The rolls were cut to various lengths to make specimens for measuring bulk ceramic properties, or pressed between glass plates to make a long rectangle, which was subsequently cut to the desired length with a razor blade.

The working time of the clay when exposed to air is short (minutes) because drying at the surface forms a crust which prevents extrusion. So, the clay must be placed in plastic bag or pressed into discs immediately. Once the discs are formed, thin discs may remain for hours between the plastic films for subsequent lamination to other pieces. For making laminated layers (such as 50/50-BZCY755/NiO//BZCY755//50/50-BZCY755/NiO trilayers), the poly-film was peeled away from the various layers, and the layers were stacked up while the clay was still slightly tacky. Typically, only a slight pressing force was required to get the layers to bond together. Once the parts were firm enough to handle, holes were punched for the gas channel inlet and outlet ports. The parts were allowed to dry in air for 24 h between wood or plaster drying blocks before final drying in an oven for one hour at 80 °C. 

### 2.3. Fabrication of Prototype Spiral Channel Discs

Circular double-spiral discs were successfully ram pressed from both the cermet formulation and the electrolyte formulation. Prototype molds were prepared by thermoplastic 3D printing from designs created using SolidWorks software. The nominal outside diameter of the molded disc was 5.2 cm and the parts were 3 mm thick with 1.5 mm deep channels. A 3D printed mold and molded parts of BZCY755 electrolyte and 50/50-BZCY755/NiO are shown in Figure 1. For fabricating enclosed channels, flat discs of the same diameter were prepared and laminated on top of the spiral channel layer prior to drying.

### 2.4. Binder Burn-Out and Sintering

During heating, and prior to sintering, it is necessary to remove all water and organics from the formed body to prevent gas build-up during sintering. Mineral oil is a purified liquid petroleum distillate with a boiling point below about 200 °C. PVA (C_4_H_6_O_2_) has a boiling point of about 112 °C. It is essential that the vapors and decomposition products ‘vent’ from the ceramic matrix without building up pressure. Since inorganics remain behind, it is important that all components of the additives either evaporate or pyrolyze completely. Materials like silicone should not be used. Thermogravimetric analysis (TGA) was used to study the temperatures at which the various components leave the matrix. It was observed that all volatile organics are absent by 400 °C. The additional weight loss up to 1000 °C is primarily due to BaCO_3_ decomposition. The low-temperature heating rate is critical to prevent damage to the specimen. Depending on the specimen cross-section, the heating rate from room temperature to 400 °C needs to be less than 100 °C/h with an additional soak interval of about one hour at 400 °C.

Sintering was carried out in a Deltech furnace with MoS_2_ heating elements, according to the schedule given in Table 1.

The best sintering results were obtained when parts were laid directly on a sintered yttria-stabilized zirconia plate. For larger parts, granules of sintered BZCY were applied between the block and the piece to aid with setter drag. Parts with 50/50-BZCY755/NiO could not be fired this way due to a setter reaction with the YSZ, and such membrane surfaces either had to be free-standing or placed on top of a packed bed of BZCY granules. Specimens for conductivity measurements were made by suspending the specimen with electrodes applied between two supports, with a sintered BZCY rod passing through a hole in one end.

### 2.5. Analysis

Post-sintered analysis was carried out to confirm the phase and microstructure. The phase purity was confirmed on a portion of crushed sample by powder X-ray diffraction using a PANalytical X’Pert X-ray diffractometer equipped with an X’Celerator detector. The lattice parameter was determined using the *Celref* software. Micrographs of fractured cross-sections were obtained on gold-coated specimens using an FEI Quanta 600i Environmental Electron Scanning Microscope (ESEM). 

The conductivity behavior was characterized by electrochemical impedance spectroscopy (EIS). First, a 1 mm thick pellet of BZCY755 with Pt electrodes painted on both sides and sintered at 1100 °C for 1 h was tested in 10% H_2_ balanced argon (100 mL/min) through a water bubbler kept at room temperature. This gas mixture is referred to as 3% moist 10% H_2_ balanced argon. The sample was heated to 750 °C and kept at that temperature until the impedance spectra did not change anymore (i.e., sample at equilibrium with the gaseous environment). Afterwards, impedance spectra were collected every 50 °C down to 300 °C, with slow cooling rates (60 °C/h) and long equilibration time (> 5 h). A second set of conductivity measurements was collected on a tri-layer sample (50/50-BZCY755/NiO//BZCY755//50/50-BZCY755/NiO) prepared as described in Section 2.3. The specimen was inserted into a fixture in a quartz tube furnace with flowing 3% moist 10% H_2_ balanced argon (50 mL/min), with platinum lead wires connected to the two cermet electrodes in 2-wire mode. The NiO/BZCY electrodes were reduced in-situ, for 24 h at 727 °C until the Ni in the electrodes was metallic and electronically percolating. The specimen was then cooled to 600 °C and analyzed by impedance spectroscopy over the course of two days until no further change was observed. The specimen was then heated to 800 °C and slowly cooled to 500 °C, with impedance spectra acquired every 25 °C. All spectra were collected with an Ivium CompactStat, with a 50 mV amplitude in the frequency range 1 MHz–10 Hz. The spectra were fitted with *ZSimpwin* software.

## 3. Results 

An example of a molded channelized disc before and after sintering is shown in Figure 2. One of the consequences of SSRS is very high shrinkage, which requires extra care in the furnace. The unfired part on the left is 5.5 cm in diameter and the fired part on the right is 3.6 cm in diameter, giving the shrinkage as ∆L/L = 34.5%.

### 3.1. X-ray Diffraction

Figure 3 displays the X-ray diffractogram recorded on a powder sample made from a BZCY755 monolithic part. BaZr_0.75_Ce_0.05_Y_0.2_O_3−d_ crystallizes as a cubic perovskite. Using the *Pm-3m* space group, the lattice parameter was estimated to be about 4.225 Å close to that of BZY20 (BaZr_0.8_Y_0.2_O_3−d_) [25].

### 3.2. Microstructure of BZCY and BZCY/Ni

A secondary electron micrograph of a fractured BZCY755 sample prepared with 0.5 wt% NiO is displayed in Figure 4 with a magnification of 5000. Well-sintered grains of 2 to 5 microns without dihedral pores were observed, typical for a low-cerium containing BZCY compound prepared by solid-state reactive sintering [22,23]. Both trans-granular and inter-granular fracture modes were present.

Cross sections of trilayer samples (BZCY755/Ni//BZCY755//BZCY755/Ni) were also investigated, combining secondary electron and back-scattered electron micrographs. In the latter case, BZCY755 phase appears as the bright phase and Ni as the grey phase. Examples of back-scattered electron micrographs are given in Figure 5 on the second sample used for conductivity measurements (i.e., in-situ NiO reduction). The interface between BZCY755/Ni and BZCY755 is continuous. BZCY755/Ni//BZCY755//BZCY755/Ni trilayers were about 0.7 mm thick, with the electrolyte thickness of about 400 microns and the BZCY755/Ni electrodes varying between 150 and 300 microns. More porous electrodes can be obtained by increasing the amount of NiO from 50 to 65 wt% and/or adding pore-formers. BZCY755/Ni electrodes with 65 wt% NiO and no pore-formers showed sufficient porosity under fuel cell and electrolysis testing [26]. It is also important to note that the sample in Figure 5 was heated to temperatures as high as 800 °C. If the reduction and testing temperatures do not exceed 700 °C, more porous Ni is observed as shown in the Appendix A. 

### 3.3. Conductivity Measurements

The impedance spectra for the BZCY755 pellet with Pt electrodes were fitted with an RL(RQ)(RQ) at high temperatures, and an L(RQ)(RQ)(RQ) at low temperatures, where R, L and Q correspond to a resistor, an inductor and a constant phase element, respectively. The impedance of the constant phase element Q is given by Z_Q_ = [Y_o_(jω)^n^]^−1^. The arcs were assigned to the corresponding processes using the pseudocapacitance (C) from Equation (1): the pseudo-capacitances were 1–2 × 10^−10^, 1–2 × 10^−8^ and about 10^−5^–10^−6^ F, which can be attributed to the bulk, grain boundaries, and electrodes respectively.
(1)$$C={\frac{\left(R_{o}Y_{o}\right)}{R_{o}}}^{1/n}.$$

An example of the spectrum collected in 3% moist 10% H_2_ balanced argon at 550 °C is shown in Figure 6a. The bulk conductivity is plotted as a function of temperature in 3% moist 10% H_2_ balanced argon in Figure 6b. As the concentration of protonic defects decreases with increasing temperature (exothermic reaction), the activation energy needs to be determined at sufficiently low temperatures (still hydrated sample): 0.51 eV in the 300–450 °C temperature range, expected value for proton conduction [27,28,29,30,31,32]. With further increase of the temperature, the protonic defect concentration decreases, leading to the shoulder in the conductivity data, before oxide ion conduction becomes non-negligible at temperatures above 700 °C [33,34]. The bulk conductivity values of BZCY755 in 3% moist 10% H_2_ balanced argon were 3.1, 2.9 and 1.6 mS.cm^−1^ at 700, 600, and 500 °C respectively, which is in the same range as values reported on BZY20 (BaZr_0.8_Y_0.2_O_3−d_) [25,35].

Fairly similar results were obtained on the BZCY755/Ni//BZCY755//BZCY755/Ni, as illustrated in Figure 7. The small difference between the conductivity of the two samples can be explained by the fact the conductivity was measured in two different test stands as well as the uncertainty in determining the thickness of the electrolyte in the trilayer specimen. It may be observed that the conductivity does not increase with temperature above 700 °C, due to decreasing proton concentration resulting from dehydration of the specimen.

## 4. Discussion

The results show that BZCY755 and BZCY755/NiO parts could be prepared using a cost- effective polymer clay process, offering flexibility for shaping that is not possible with traditional processing methods. Highly dense electrolyte specimens were prepared, with the desired microstructure, i.e., well-sintered grains without dihedral pores. In addition, preparation of layered materials is possible, using lamination while the joined members are in the tacky stage. 

Figure 8 summarizes the various shapes of samples and geometries that have been prepared in this study: flat channelized 50/50-BZCY755/NiO and BZCY755 substrates, BZCY755 rods, laminated 50/50-BZCY755/NiO//BZCY//50/50-BZCY755/NiO trilayers and L-shaped BZCY755 specimens. For the last sample type, two pieces of BZCY755 were folded over the corner of a block of wood while the clay was still soft. Note that the zirconia media balls in Figure 8 are there to keep the samples from flopping over. Figure 9 displays a secondary electron micrograph of a laminated flat channelized BZCY755 part (top) onto a 50/50-BZCY755/NiO pellet (bottom). The channels survived the lamination and sintering, with a visibly good adhesion between the two parts.

One limitation of the process is the minimal thickness that can be obtained, around 200 microns. For high-performance devices, the electrolyte membrane should be as thin as possible, while still being gas-tight. Half cells (thin BZCY755 onto 50/50-BZCY755/NiO) were prepared using a green channelized 50/50-BZCY755/NiO piece similar to the right specimen in Figure 1 and painting a BZCY755 ink (prepared with 3 wt% ethylcellulose in terpineol) on the flat side. The two layers were then co-sintered with the sintering schedule given in Table 1. A 50–70 micron thick BZCY755 layer was formed onto the BZCY755/NiO channelized support with a good interface (Figure 10).

## 5. Conclusions

A process for making thick BZCY proton conductors by solid-state reactive sintering from polymer clay was described. This simple process is suitable for making a wide variety of parts needed to fabricate hydrogen separation membrane reactors. It is possible to make discs and plates for conductivity measurements, rods for measuring mechanical properties, button cells, electrolyte-supported cells, and channelized substrates for gas manifolding by molding the plastic clay body in the ‘green’ state. Most importantly, the sintered ceramic bodies made from polymer clay were shown to have the microstructure and electronic properties required for electrochemical devices fabricated from these materials. Future work will focus on electrode development, and the mechanical aspects and stability of these samples. Furthermore, preparing ceramic components from polymer clay is similar to the processes used for making dinner plates and sanitary ware at an enormous scale. By applying many of the industrial processes employed in this industry, it is easy to imagine polymer clay becoming the preferred method for scaling BZCY protonic ceramics to the level of thousands of square meters that will ultimately be required for commercialization of the technology.

## Figures and Tables

**Figure 1 membranes-09-00130-f001:**
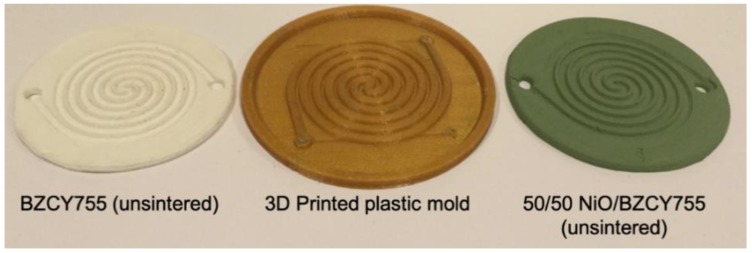
The 3D printed mold (**center**), molded parts BZCY755 (**left**) BZCY755/NiO (**right**).

**Figure 2 membranes-09-00130-f002:**
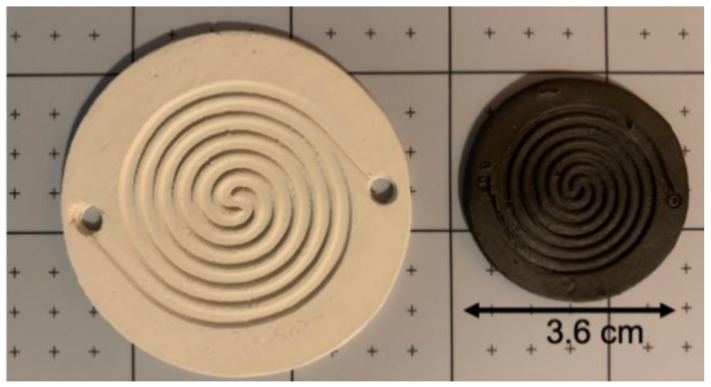
BZCY755 molded spiral channel green (**left**), fired (**right**).

**Figure 3 membranes-09-00130-f003:**
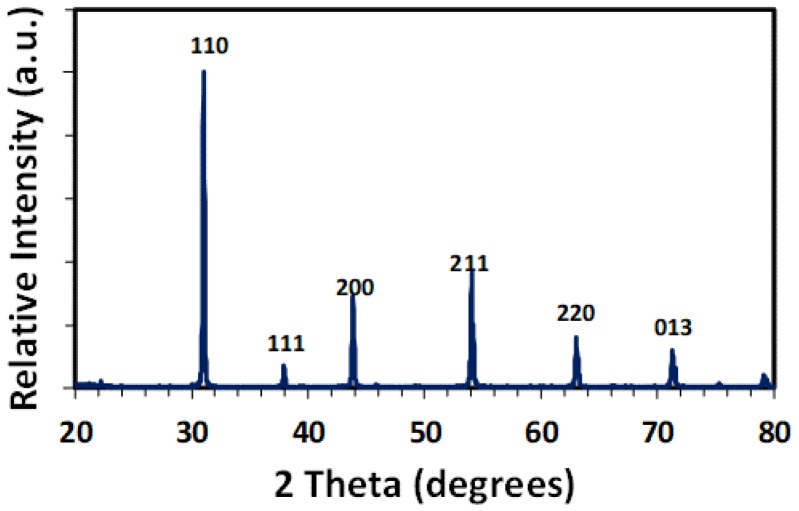
X-Ray diffraction pattern recorded on a BZCY755 sintered part.

**Figure 4 membranes-09-00130-f004:**
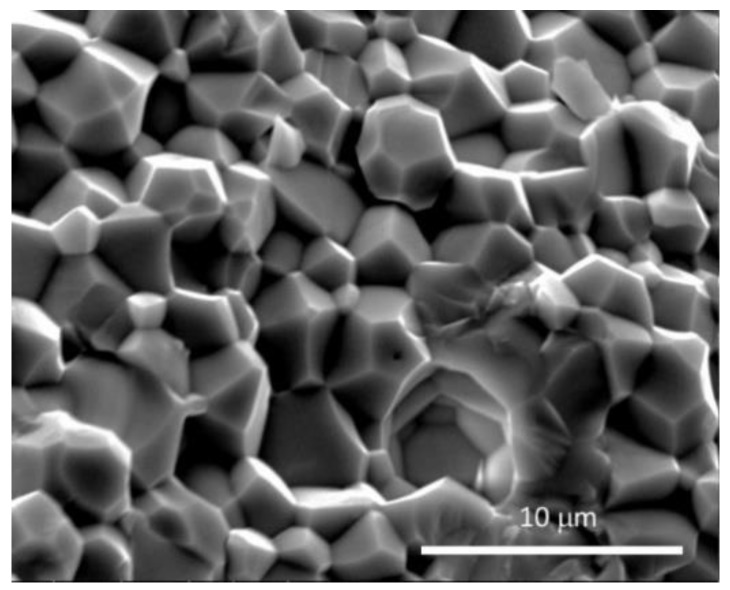
Secondary electron micrograph of sintered BZCY755 fractured cross section (5000×).

**Figure 5 membranes-09-00130-f005:**
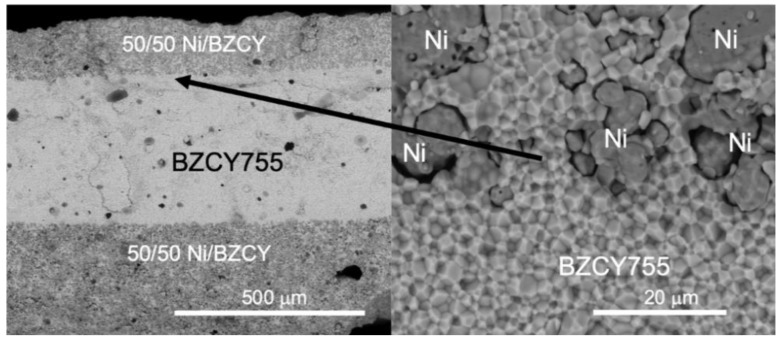
Back-scattered electron micrographs of sintered and reduced BZCY755/Ni//BZCY755//BZCY755/Ni fractured cross section.

**Figure 6 membranes-09-00130-f006:**
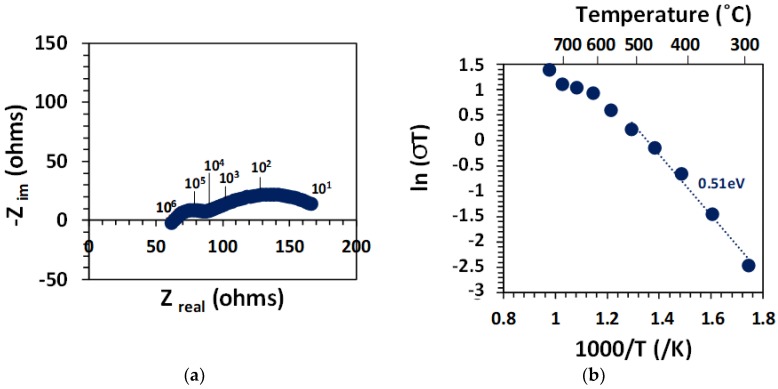
(**a**) Nyquist plot of the spectrum at 550 °C. (**b**) Bulk conductivity on a BZCY755 specimen, with Pt electrodes in 3% moist 10% H_2_ balanced argon.

**Figure 7 membranes-09-00130-f007:**
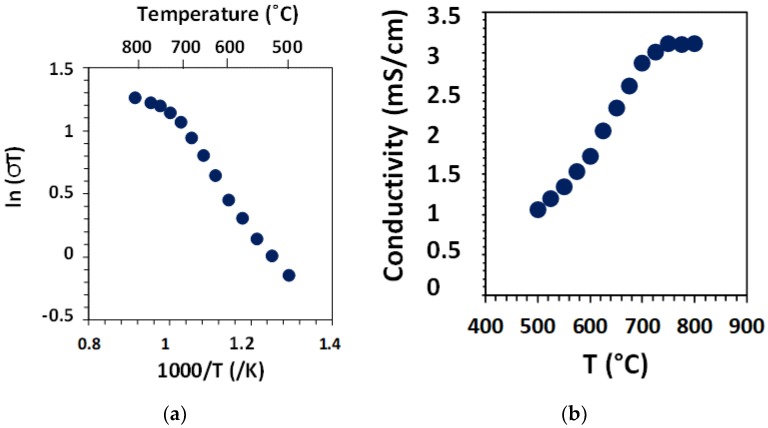
Bulk conductivity on a BZCY755 specimen (**a**), with BZCY755/Ni electrodes (**b**) in 3% moist 10% H_2_ balanced argon.

**Figure 8 membranes-09-00130-f008:**
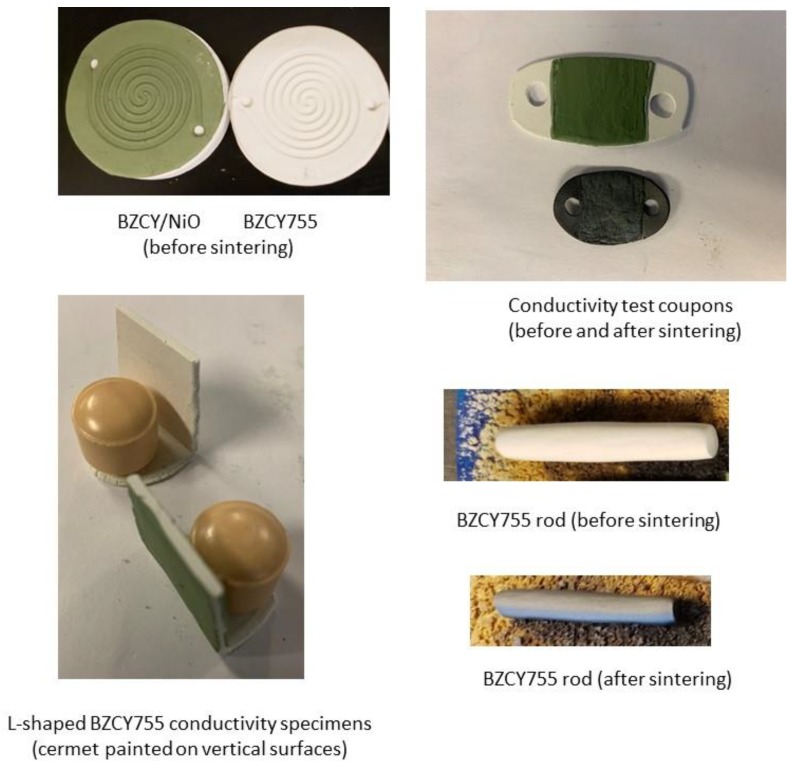
Examples of specimens prepared by the polymer clay process.

**Figure 9 membranes-09-00130-f009:**
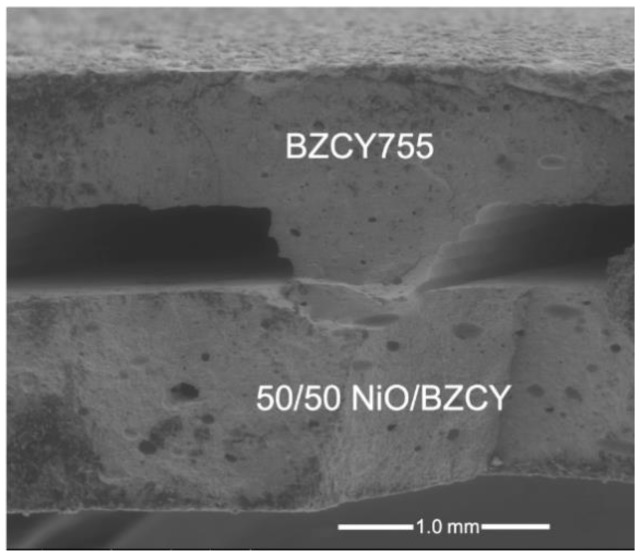
Secondary electron micrograph of laminated microstructure of a channelized substrate (unreduced) in cross-section.

**Figure 10 membranes-09-00130-f010:**
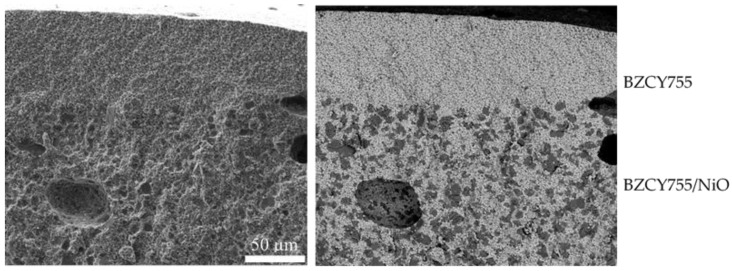
Secondary electron (**left**) and back-scattered electron (**right**) micrographs of a thin BZCY755 film painted and co-fired onto the flat side of the polymer clay channelized 50/50-BZCY755/NiO sample (fractured cross section). (BZCY: light phase, NiO: grey phase).

**Table 1 membranes-09-00130-t001:** Sintering schedule.

Start Temp (°C)	End Temp (°C)	Rate (°C/h)	Soak (h)
RT	200	60	-
200	600	20	-
600	600	-	2
600	1550	180	-
1550	1550	-	5
1550	RT	180	-

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
