# Peer review of "Channelized Substrates Made from BaZr0.75Ce0.05Y0.2O3−d Proton-Conducting Ceramic Polymer Clay"

_membranes, 2019, doi:10.3390/membranes9100130_

Round 1

Reviewer 1 Report

The manuscript “Channelized substrates made from ceramic polymer clay” presents a processing route for potential large scale proton conducting membrane fabrication for hydrogen separation. The work is interesting and deals with the issue of difficult processing and scalability of these materials.

In the case of Ni cermet electrodes, one would expect the presence of porosity as a requirement for practical use and this aspect seems to be overlooked in this work. It would be expected the formation of a porous structure upon NiO reduction as well as Ni percolation and this does not seem evident (from Figure 5). One drawback of this method, also acknowledged by the authors is the scale of microstructural control, as thin components and pores with controllable sizes (in the micron range) and orientation are desirable for practical application.

A couple of comments: the overall reporting style is a bit unusual, starting with the Materials and Methods section where the amounts of reagents are reported in wt% as related to “stoichiometry”. The section continues with quite empirical terms such as “a few drops”. In the Conductivity measurements section, pseudo capacitance values reported for Q in the electrochemical testing are not very useful if unconverted. The temperature intervals are defined as low and high with no mention of the limit where bulk cannot be observed. The last point above 700 oC in Fig 6b looks out of trend and it would have been useful if the conductivity or resistance of Ni-BZCY755 would have been discussed. I would recommend the conductivity units to be the same in the graphs 6b and 7 to prove the point. No reference to the mechanical aspects and stability of these structures (i.e bi-layer) is made.  

Author Response

In the case of Ni cermet electrodes, one would expect the presence of porosity as a requirement for practical use and this aspect seems to be overlooked in this work. It would be expected the formation of a porous structure upon NiO reduction as well as Ni percolation and this does not seem evident (from Figure 5).

The reviewer highlights an important point. In this work, 50-50 wt. % electrode were prepared as a proof of concept. Higher NiO amount (65 wt. %) should be used to obtain more porosity. In former studies, it was proved that the porosity from the NiO reduction is sufficient for hydrogen diffusion through the electrolyte. Another important point is that the nickel coarsens at high temperatures. When the temperature does not exceed 700C, more porous Ni is observed. This is now discussed on page 7 and a figure has been added as supplementary information.

One drawback of this method, also acknowledged by the authors is the scale of microstructural control, as thin components and pores with controllable sizes (in the micron range) and orientation are desirable for practical application.

No changes were made.

A couple of comments: the overall reporting style is a bit unusual, starting with the Materials and Methods section where the amounts of reagents are reported in wt% as related to “stoichiometry”. The section continues with quite empirical terms such as “a few drops”.

The empirical terms have been replaced.

In the Conductivity measurements section, pseudo capacitance values reported for Q in the electrochemical testing are not very useful if unconverted.

The pseudo capacitance of the Q element are converted using the formula: C = ( Qo * R )(1/n) / R

The temperature intervals are defined as low and high with no mention of the limit where bulk cannot be observed.

The low and high temperature regions were defined to determine the activation energy. The activation energy should not be determined when the sample dehydrates, this is now clarified in page 7 of the manuscript.

The last point above 700 oC in Fig 6b looks out of trend and it would have been useful if the conductivity or resistance of Ni-BZCY755 would have been discussed.

The frequency range was not sufficient to analyze in details the conductivity of BZCY/Ni electrodes. The focus of this paper was the electrolyte conductivity to prove that expected values were obtained.

I would recommend the conductivity units to be the same in the graphs 6b and 7 to prove the point.

Figure 7 has been modified. The values initially plotted in figure 6 were corrected from the Bruggeman method (Bruggeman, D.A.G. Berechnung verschiedener physikalischer Konstanten von heterogenen Substanzen. I. Dielektrizitätskonstanten und Leitfähigkeiten der Mischkörper aus isotropen Substanzen. Ann. Phys. 1935, 416, 636–664) to estimate the conductivity of the fully dense sample. This is now changed to the actual measured values, corresponding to values given of about 3 mS/cm at 600C.

No reference to the mechanical aspects and stability of these structures (i.e bi-layer) is made.

This will be part of future work. This is added in the conclusions.

Reviewer 2 Report

This is a very readable and interesting paper describing how different shapes of ceramic proton-conducting membrane material may be processed employing a ceramic polymer clay. The paper will be of considerable interest to both ceramists and the solid-state ionics community.

The authors may like to consider the following minor points prior to publication.

The title is very general, whereas the actual work focusses on a very specific material, BZCY55. Could the title reflect better that this process is applied here to a ceramic proton conductor? Moreover, it seems that solid state reaction sintering is required for the processing to be effective, thereby limiting itself to a discrete number of systems, such as the present one.

L28. “This means that…..in some predictable way.” This sentence is rather cumbersome and difficult to follow.

L158…”thin discs may can remain”…should read “..thin discs may remain…”

L184. “Thermographic analysis”. Do you not mean “Thermogravimetric analysis”?

L196. “…either had to free-standing…” should read “…either had to be free-standing…”

Fig. 5. The Ni in the Ni-BZCY55 electrodes seems poorly dispersed and the porosity appears low. Any comment on the quality of the electrodes and if they are affecting the electrical measurements?

Fig. 6(b) and 7 are both conductivity vs temperature plots but in different formats. It would be more helpful and coherent to have the same representation in the two plots.

Author Response

The title is very general, whereas the actual work focusses on a very specific material, BZCY55. Could the title reflect better that this process is applied here to a ceramic proton conductor? Moreover, it seems that solid state reaction sintering is required for the processing to be effective, thereby limiting itself to a discrete number of systems, such as the present one.

The title has been modified to ‘Channelized substrates made from BaZr0.75Ce0.05Y0.2O3-d proton-conducting ceramic polymer clay’

L28. “This means that…..in some predictable way.” This sentence is rather cumbersome and difficult to follow.

It has been rewritten.

L158…”thin discs may can remain”…should read “..thin discs may remain…”

It has been corrected.

L184. “Thermographic analysis”. Do you not mean “Thermogravimetric analysis”?

It has been corrected.

L196. “…either had to free-standing…” should read “…either had to be free-standing…”

It has been corrected.

Fig. 5. The Ni in the Ni-BZCY55 electrodes seems poorly dispersed and the porosity appears low. Any comment on the quality of the electrodes and if they are affecting the electrical measurements?

For the conductivity measurements, the electrode contribution was very negligible compared to the bulk one so it was not an issue. For better performing electrode, higher NiO amount (typically 65 wt.%) should be used. A paragraph has been added on page 7, as well as a supplementary micrograph.

Fig. 6(b) and 7 are both conductivity vs temperature plots but in different formats. It would be more helpful and coherent to have the same representation in the two plots.

Figure 7 has been modified. The values initially plotted in figure 6 were corrected from the Bruggeman method (Bruggeman, D.A.G. Berechnung verschiedener physikalischer Konstanten von heterogenen Substanzen. I. Dielektrizitätskonstanten und Leitfähigkeiten der Mischkörper aus isotropen Substanzen. Ann. Phys. 1935, 416, 636–664) to estimate the conductivity of the fully dense sample. This is now changed to the actual measured values, corresponding to values given of about 3 mS/cm at 600C.

Round 2

Reviewer 1 Report

In the revised version of the manuscript, although an explanation for the lack of porosity observed in Figure 5 is provided, it is quite unlikely that no apparent porosity (nor percolation) could be present even at this percentage of NiO and even after heating at 800 C. The new picture looks very different that the one initially attached in the text. In the Analysis part, the reduction conditions are mentioned as 24 h at 727 C. It looks like the authors omitted to introduce the converted pseudo-capacitance values of the equivalent circuits. The new reference is inaccurate and the legend of the supplementary figure does not describe the cross section of 50/50–BZCY755/Ni//BZCY755//50/50–BZCY755/Ni sample but the electrode itself. In the revised version of Figure 7 the top temperature scale looks wrong and the conductivity values at low temperatures quite different that the ones reported in Figure 6b. As no impedance spectrum is provided or details on the analysis of responses (possible errors in estimation, complex response, etc), a discussion might be necessary on this observation.

Author Response

The authors acknowledge the reviewer for the additional comments. The responses are given below and are highlighted in pink in the revised version of the manuscript.

(1) In the revised version of the manuscript, although an explanation for the lack of porosity observed in Figure 5 is provided, it is quite unlikely that no apparent porosity (nor percolation) could be present even at this percentage of NiO and even after heating at 800 C. The new picture looks very different that the one initially attached in the text. In the Analysis part, the reduction conditions are mentioned as 24 h at 727 C.

The sample from Figure 5 was indeed reduced at 727C but was then heated at 800C, hydrated and conductivity measurements were performed from 800C to 500C. That is why the authors wrote ‘It is also important to note that the sample in Figure 5 was heated to temperatures as high as 800 ºC’ on page 7. That sample was at least one day at temperatures above 700C.

(2) It looks like the authors omitted to introduce the converted pseudo-capacitance values of the equivalent circuits.

The authors didn’t think it was necessary to add the equation for the pseudocapacitances as it can easily be found in the literature. It is now added on page 7.

(3) The new reference is inaccurate

This is now corrected.

(4) The legend of the supplementary figure does not describe the cross section of 50/50–BZCY755/Ni//BZCY755//50/50–BZCY755/Ni sample but the electrode itself.

The caption of supplementary figure has been modified.

(5) In the revised version of Figure 7 the top temperature scale looks wrong and the conductivity values at low temperatures quite different that the ones reported in Figure 6b. As no impedance spectrum is provided or details on the analysis of responses (possible errors in estimation, complex response, etc), a discussion might be necessary on this observation

The authors are really thankful that the reviewer caught this mistake. The top axis was correct but the bottom axis was wrong. The measurement for the trilayer was performed only from 800-500C, as described in the experimental section. The figure is now corrected. After this correction, the difference between the two samples is much smaller. A sentence has been added on page 8.
